# Estimate the Earliest Phenophase for Garlic Mapping Using Time Series Landsat 8/9 Images

Yan Guo [1], Haoming Xia [1,2,3,4,*], Xiaoyang Zhao [1], Longxin Qiao [1] and Yaochen Qin [1,3,4]

1 College of Geography and Environmental Science, Henan University, Kaifeng 475004, China
2 Henan Key Laboratory of Integrated Air Pollution Control and Ecological Security, Kaifeng 475004, China
3 Key Laboratory of Geospatial Technology for the Middle and Lower Yellow River Regions, Henan University, Ministry of Education, Kaifeng 475004, China
4 Key Research Institute of Yellow River Civilization and Sustainable Development Collaborative Innovation Center on Yellow River Civilization Jointly Built by Henan Province and Ministry of Education, Henan University, Kaifeng 475004, China
* Correspondence: xiahm@vip.henu.edu.cn

**Abstract:** Garlic is the major economic crop in China. Timely and accurate identification and mapping of garlic are significant for garlic yield prediction and garlic market management. Previous studies on garlic mapping were mainly based on all observations of the entire growing season, so the resulting maps have a hysteresis. Here, we determined the optimal identification strategy and the earliest identifiable phenophase for garlic based on all available Landsat 8/9 time series imagery in Google Earth Engine. Specifically, we evaluated the performance of different vegetation indices for each phenophase to determine the optimal classification metrics for garlic. Secondly, we identified garlic using random forest algorithm and classification metrics of different time series lengths. Finally, we determined the earliest identifiable phenophase of garlic and generated an early-season garlic distribution map. Garlic could be identified as early as March (bud differentiation period) with an F1 of 0.91. Our study demonstrates the differences in the performance of vegetation indices at different phenophases, and these differences provide a new idea for mapping crops. The generated early-season garlic distribution map provides timely data support for various stakeholders.

**Keywords:** early-season; garlic identification; phenology; Landsat 8/9; separability

## 1. Introduction

Garlic is one of the primary economic crops (garlic, peanut, rape, sugar beet, sugarcane, cotton, etc.) in China. According to the Food and Agriculture Organization (FAO) of the United Nations, China's garlic production reached 23.3 million tons in 2019, making China the largest garlic growing and producing region in the world (http://www.fao.org/faostat, last accessed 15 January 2022). Timely and accurate mapping of garlic distribution is an important way to obtain the dynamics of garlic planting and is of great significance for the planning and management of the garlic industry.

Traditionally, the information on garlic planting has been obtained mainly based on field surveys. This method is not only vulnerable to subjective factors [1], but also has a long cycle and is labor-intensive and time-consuming [2,3]. With the development of cloud computing platforms and the opening of massive satellite data archives [4], satellite remote sensing has become a feasible method for crop type mapping and has been widely used in many fine mapping fields, such as urban land mapping [5], water surface area change [6], forest degradation [7], cropland classification [8], and wetland classification [9].

Given the highly overlapping growth cycles and relatively similar phenological cycles of garlic to other winter crops (e.g., winter wheat, rapeseed), it is difficult to separate garlic from other winter crops using a single vegetation index. Existing studies on garlic mapping have focused on the use of very-high-resolution commercial satellite imagery (e.g.,

Worldview-2) [10] or ultra-high-resolution imagery (e.g., unmanned aerial vehicle) [11]. However, the high cost of commercial satellites and the limited scope of UAV make it difficult to obtain regional garlic planting. In addition, these maps use observations throughout the lifecycle, so garlic distribution is usually obtained after harvest, which is too late for government policymakers to predict garlic planting area and yield [12], market traders to regulate garlic prices [13], and farmers to implement planting decisions. Therefore, there is a need to monitor garlic distribution early or in the middle of the growing season, to provide timely data support for the scientific management of the garlic industry and to provide an assessment basis for insurance in extreme weather events and natural disaster regions. In addition, obtaining a garlic distribution map in the early or middle growing season means using fewer data, which not only eases data processing but also may address the lack of imagery in some regions.

Methodologically, crop mapping based on remote sensing data can usually be divided into (1) visual interpretation [14,15], (2) object-based [16,17], and (3) pixel-based crop classification methods [16–20]. Visual interpretation is based on the manual interpretation of images based on expert experience, which is a costly method and is susceptible to subjective factors. Object-based classification algorithms include hierarchical image segmentation software [21], etc. Pixel-based classification algorithms include random forest algorithms [22,23], support vector machines [24,25], decision tree algorithms [26], phenological methods [18,27], etc. Such approaches are based on spectral features, vegetation indices, or texture features derived from single-date or multi-date images, applying supervised or unsupervised classification methods to generate crop maps. As a popular classification algorithm, random forest (RF) is a combination of multiple decision trees. Each decision tree obtains a training set from ground reference data and assigns a class label to the data after training. Ultimately, the class of each pixel is determined by majority voting to fuse the results of all decision trees [28].

However, the data are often irregular due to overlapping satellite orbits and the influence of clouds [29], which usually need to be processed by smoothing and function-fitting algorithms before further analysis [30]. Moreover, the use of filters to smooth the time series can remove noise while maintaining the original trend of the curve, and does not affect the reproducibility of the experiment. Commonly used filtering methods include Savitzky–Golay filter, least-square fits to asymmetric Gaussian functions [31], double logistic functions [32], Whittaker smoother [33], wavelet transforms [34], the best index slope extraction [35], etc. Here, we use the Savitzky–Golay filter, which is based on local polynomial least-squares fitting in the time domain and can keep the authenticity of the time series as much as possible [36].

Efforts have been made to generate early crop products using satellite observations, such as rice [37], corn [38,39], soybeans [40], and winter crops [41]. However, these studies rely on coarse spatial resolution data, resulting in parcel-based fields in numerous agricultural landscapes that cannot be resolved [42]. These do not migrate for use in Asia and Africa, where parcel-level cropland (normally less than 0.2 ha) is widely available [43]. Therefore, it is necessary to map crops of complex planting areas based on images with higher spatial resolution.

The Landsat series of satellites, with a spatial resolution of 30 m, is considered one of the most reliable satellite products and is a vital data source for crop identification. With the launch of the latest Landsat 9, higher radiometric precision and higher signal-to-noise ratio images are available, and their spectral, spatial, and radiometric measurements are consistent with those of Landsat 8. More importantly, the combination of Landsat 8 and Landsat 9 improves the revisit cycle of the satellites, which provides important support for obtaining key crop phenophase.

Considering the actual demands of governments, market traders, and farmers for timely and accurate information on garlic cultivation, three objectives were proposed in this study: (1) to evaluate the performance of multiple vegetation indices and determine the optimal separable metrics for garlic mapping in each phenophase; (2) to identify the

earliest phenophase of garlic mapping based on different length classification metrics time series; and (3) to develop an automated mapping framework to map early-season garlic distribution.

## 2. Materials and Methods

### 2.1. Study Area

Qi County lies in the east of Henan Province (Figure 1a). It has a temperate continental monsoon climate. The total terrain is high in the northwest and low in the southeast (Figure 1b). The main crops are garlic and winter wheat. Qi County is one of the six major garlic-producing regions in China. According to statistics, garlic production reached 84.8 tons in 2019 (https://data.cnki.net/Yearbook, last accessed 15 January 2022), ranking first at the county level in China.

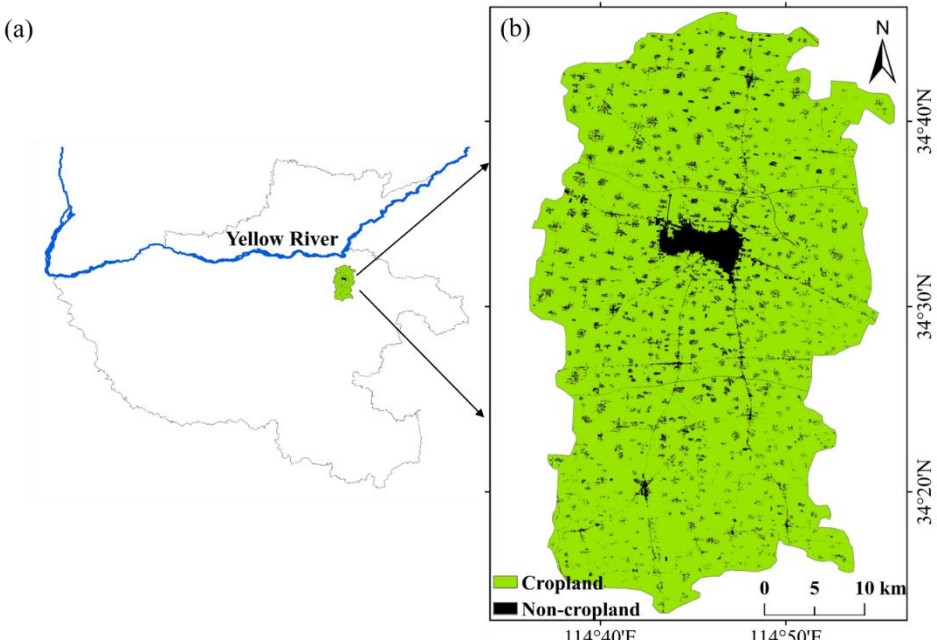

**Figure 1.** (**a**) Location of Qi County, Henan Province. (**b**) Spatial pattern of the cropland.

### 2.2. Datasets and Preprocessing

2.2.1. Landsat 8/9 Images

Landsat 8/9 were launched in February 2013 and December 2020, respectively, carrying Operational Land Image (OLI) and Operational Land Image-2 (OLI-2) with a spatial resolution of 30 m and a revisit period of 16 days. Landsat 8 images and Landsat 9 images correspond to "LANDSAT/LC08/C02/T1_L2" and "LANDSAT/LC09/C02/T1_L2" in the Google Earth Engine (GEE) cloud platform, respectively. All of these data are surface reflection data, which are the physically based normalization of the image values throughout the dates regardless of the different atmospheric conditions [44].

2.2.2. Ground Reference Data

The ground reference data of different crop types are critical for verifying the crop classification algorithm. Ground reference data were collected through the following three aspects. First, ground training and validation samples were collected based on the very-high-spatial-resolution images in Google Earth, which clearly show different ground cover types. Second, three survey routes were designed, and fieldwork was conducted from March 2022 to June 2022 to collect georeferenced field photos. These field photos included different crop types, such as garlic and winter wheat. Third, multiple multispectral images with a spatial resolution of up to 0.1 m were obtained by UAV during the fieldwork. In these images, the garlic fields are significantly lighter colored than

the wheat fields, which made garlic easily distinguishable from wheat. With the images obtained from the above three aspects, 531 garlic samples and 458 non-garlic crop samples were collected (Figure 2), including 90 garlic training samples, 82 non-garlic crop training samples, 441 garlic validation samples, and 376 non-garlic crop validation samples.

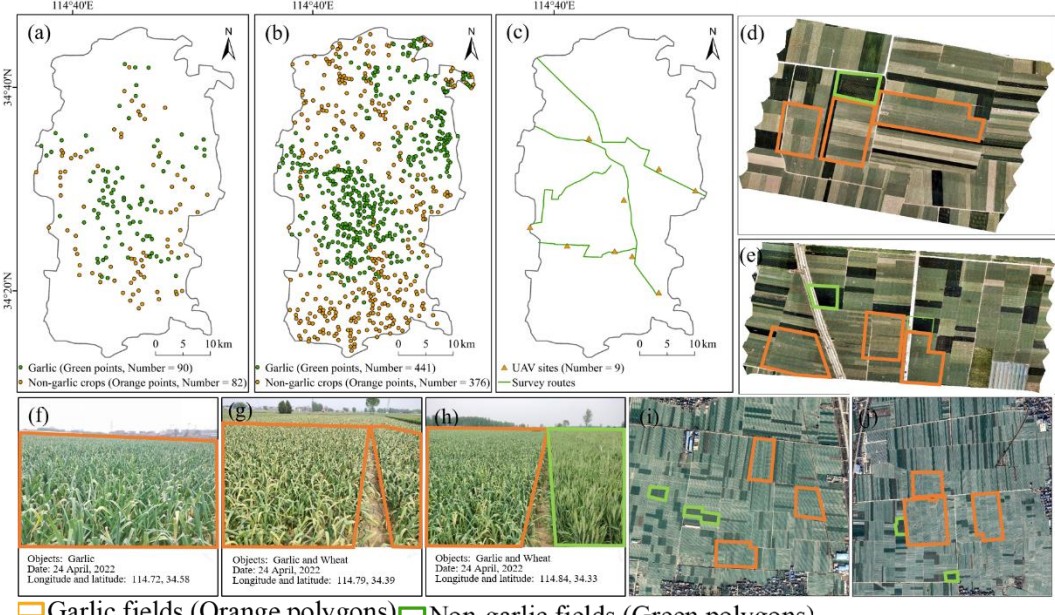

**Figure 2.** (**a**) Distribution of training samples for garlic and non-garlic crops. (**b**) Distribution of validation samples, including polygons of garlic and non-garlic crops. (**c**) Distribution of UAV sites and survey routes. (**d**,**e**) UAV images. (**f**–**h**) Ground observations from field camera. (**i**,**j**) Very-high-spatial-resolution images from Google Earth.

### 2.2.3. Land Cover Data

Cropland data from the GlobeLand30 land cover product were used to mask the study area to delimit the cropland extent. The product is based on the 30 m spatial resolution multispectral images (including Landsat TM, ETM+, OLI, HJ-1) and the 16 m spatial resolution GF-1 multispectral images. The overall accuracy of the product can reach 85.72% with a kappa coefficient of 0.82. The map can be freely accessed (http://www.globallandcover.com/, last accessed 15 January 2022).

### 2.3. Methods

Figure 3 shows the algorithm framework for determining the earliest identifiable phenophase for garlic and producing the distribution map of garlic in 2022. First, based on all available Landsat 8/9 imagery from November 2021 to June 2022, we quantitatively evaluated the performance of multiple vegetation indices to determine the optimal classification metrics for each phenophase. Second, the optimal classification strategy was generated based on the optimal classification metrics. Third, input variables of different lengths were constructed in steps of one phenophase and input random forest (RF) classifier. Finally, the earliest identifiable phenophase was determined and an early-season map of garlic with 30 m spatial resolution was generated.

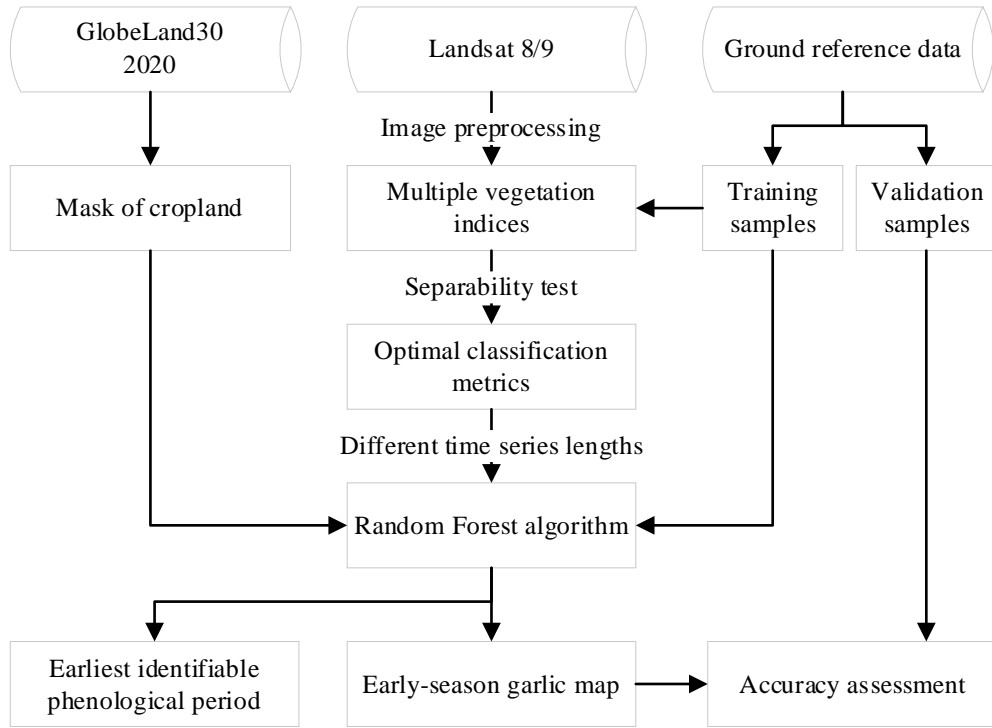

**Figure 3.** The algorithm framework for identifying and mapping garlic distribution.

### 2.3.1. Vegetation Indices Calculation

Spectral indices that are sensitive to vegetation greenness [45] can be used to capture the physical differences of different land cover types [46] and to characterize the growth curves of different crop types [47]. In this study, we calculated multiple indices presented in Table 1, where $\rho_{red}$, $\rho_{green}$, $\rho_{blue}$, $\rho_{nir}$, and $\rho_{swir}$ represent the red, green, blue, NIR, and SWIR band, respectively.

**Table 1.** The indices used in this study.

| Index | Formulas | Implication | Reference |
|---|---|---|---|
| Green Chromatic Coordinate (GCC) | $GCC = \frac{\rho_{green}}{\rho_{red} + \rho_{green} + \rho_{blue}}$ | GCC is originally designed for use with a digital RGB camera to measure wheat cover. | [48] |
| Green Leaf Index (GLI) | $GLI = \frac{2\rho_{green} - \rho_{red} - \rho_{blue}}{2\rho_{green} + \rho_{red} + \rho_{blue}}$ | GLI is sensitive to green leaves and can be used to measure leaf chlorophyll content. | [49] |
| Normalized Green–Red Difference Index (NGRDI) | $NGRDI = \frac{\rho_{green} - \rho_{red}}{\rho_{green} + \rho_{red}}$ | NGRDI is similar to NDVI, but uses the green band instead of the NIR band. | [50] |
| Normalized Green–Blue Difference Index (NGBDI) | $NGBDI = \frac{\rho_{green} - \rho_{blue}}{\rho_{green} + \rho_{blue}}$ | NGBDI is based on NGRDI, using the blue band instead of the red band. | [50] |
| Enhanced Vegetation Index (EVI) | $EVI = 2.5 \times \frac{\rho_{nir} - \rho_{red}}{\rho_{nir} + 6.0\rho_{red} - 7.5\rho_{blue} + 1}$ | EVI is highly related to leaf area index and chlorophyll in the canopy. | [51] |
| Normalized Difference Vegetation Index (NDVI) | $NDVI = \frac{\rho_{nir} - \rho_{red}}{\rho_{nir} + \rho_{red}}$ | NDVI is highly related to leaf area index and chlorophyll in the canopy. | [52] |
| Green Normalized Difference Vegetation Index (GNDVI) | $GNDVI = \frac{\rho_{nir} - \rho_{green}}{\rho_{nir} + \rho_{green}}$ | GNDVI is more sensitive to chlorophyll concentration than NDVI. | [53] |
| Optimized Soil-Adjusted Vegetation Index (OSAVI) | $OSAVI = \frac{\rho_{nir} - \rho_{red}}{\rho_{nir} + \rho_{red} + 0.16}$ | OSAV is effective in identifying chlorophyll content of plants in the early stage of growth. | [54] |
| Modified Normalized Difference Water Index (MNDWI) | $MNDWI = \frac{\rho_{green} - \rho_{swir}}{\rho_{green} + \rho_{swir}}$ | MNDWI uses green and SWIR bands to enhance open water features. | [55] |

### 2.3.2. Separability Test

The separability test reflects the distinguishing abilities of the metrics in detecting two different categories. We used separability test to evaluate the potential of the above vegetation indices in distinguishing garlic (Equation (1)) [56]. M < 1 means that the two categories overlap significantly and the classification ability is poor; M > 1 means that the two categories are relatively easy to separate [57–59]. In this study, garlic and non-garlic were tested as two categories. For each phenophase, we selected three vegetation indexes with M > 1 as classification metric, which means that the classification indexes of different phenophases were different.

$$M = \frac{|\mu_1 - \mu_2|}{\sigma_1 + \sigma_2} \tag{1}$$

where $\mu_1$ and $\mu_2$ represent the mean value for class 1 and class 2, and $\sigma_1$ and $\sigma_2$ represent the standard deviation of value for class 1 and class 2, respectively.

### 2.3.3. Time Series Construction

Compositing images at a regular interval can reduce the impact of clouds and uneven observations in time [60,61]. Therefore, the VIs time series were constructed at a 10 d interval. The maximum value of all good-quality observations within a 10 d period was taken as the observation value of the 10 d period. When there was no good-quality observation in a 10 d period due to clouds, cloud shadow, and snow effects, the linear interpolation method was used to fill the data gap, which can be considered as a gap-filling reconstruction strategy recovering missing information [62]. This is one of the most well-known and used time series reconstruction methods based on a sliding temporal window [62]. The interpolated data depend on good-quality observations before and after the 10 d interval [63]. However, even after the above steps, extremely high or low outliers may still appear in the time series, which is usually caused by cloud containment, atmospheric variability, and bidirectional effects. Therefore, the Savitzky–Golay filter was used to smooth the time series with a moving window of size 9 and a filter order of 2 [64].

Garlic is usually sown in early November and enters the seedling period after a germination period of more than ten days. Seedling period usually lasts until March of the following year and then enters the bud differentiation period. The buds continue to differentiate until the formation of garlic bolt enters the elongation period. In early May, garlic enters the expansion period, and the bulb continues to expand until the final harvest. To explore the earliest recognizable phenophase of garlic, we took the germination period (1 November 2021) as the fixed starting period, and the end date gradually lengthens with the phenophase as the step, and successively generated time series of different lengths as the subsequent input variables.

### 2.3.4. Classifier Setup and Training

As an algorithm for aggregating various decision trees [65], RF are highly robust and efficient. We used the RF algorithm on GEE, where minleafpopulation was set to 10, numberoftrees was set to 100, and other parameters used default values [66]. In this study, we used the random forest classification algorithm provided by the GEE platform. The training samples were input to the classifier in a ratio of 7:3, while the time series of multiple indices obtained in Section 2.3.3 were used as test data to obtain classification results based on time series of different lengths. Later, the classification results were validated for accuracy using validation samples and confusion matrices to obtain user accuracy (UA) and producer accuracy (PA), which were further used to calculate F1. As the length of the time series of the input variables increased, the F1 continually increased. The earliest identifiable phenophase of garlic was defined as the threshold at which F1 first reached 0.9, since it tended to stabilize when F1 reached 0.9 (Equation (2)).

$$F1 = 2 \times \frac{UA \times PA}{UA + PA} \tag{2}$$

2.3.5. Map Generation and Accuracy Assessment

Early-season garlic distribution maps were derived from a binary classifier, where pixels were divided into garlic fields and non-garlic fields. The accuracy of the maps was assessed using the validation samples described in Section 2.2.2 and confusion matrix method [64]. Similar to the determination of the earliest recognizable phenophase, the early-season garlic distribution map was defined as the F1 first exceeding 0.9.

Specifically, the accuracy assessment steps were as follows. Firstly, the validation samples obtained in Section 2.2.2 were loaded into Google Earth, and the vector boundary of homologous ground reference point was plotted according to the actual size and shape of cropland. Secondly, the attributes of the vector polygons were labeled by visual interpretation and field survey data. Thirdly, these vector polygons with attribute information were converted into raster data with a spatial resolution of 30 m. These raster data were used as the validation samples to construct the confusion matrix with the classification results.

## 3. Results

### 3.1. Optimal Classification Metrics Determination

Figure 4 shows the results for the separability of each vegetation index. The M values of vegetation indices varied during different phenophases. It should be emphasized that the separability test was based on the original values of all garlic and non-garlic training samples (only the cloud is removed). Therefore, the curve is not continuous. Based on the M value and observation frequency, GLI, NGRDI, and MNDWI were selected at germination period, GLI, NGBDI, and MNDWI were selected at seedling period, NDVI, GNDVI, and OSAVI were selected at bud differentiation period, NDVI, GNDVI, and MNDWI were selected at elongation period, and NGRDI, EVI, and NDVI were selected at expansion period.

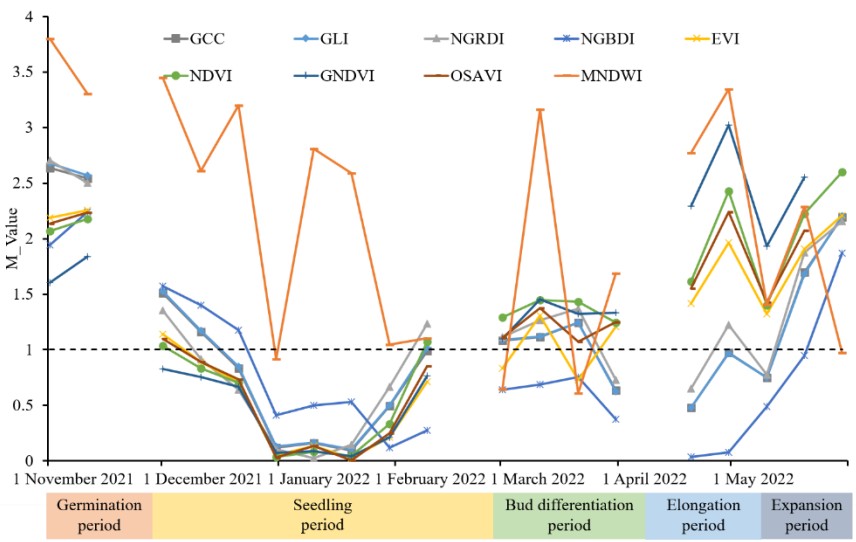

**Figure 4.** Separability results of classification metrics.

### 3.2. Earliest Identifiable Phenophase Determination

As the length of the time series of the input classification metrics continually increased, F1 kept increasing, as shown in Figure 5. The time series of input classifiers were processed as described in Section 2.3.3. The F1 obtained from three synthetic images of germination period was 0.75, from 13 synthetic images of germination period and seedling period was 0.83, from 17 synthetic images of germination period, seedling period, and bud differentiation period was 0.91, from 20 synthetic images of germination period to elongation period was 0.92, and from 22 synthetic images of germination period to expansion period was 0.93. Therefore, the earliest identifiable phenophase of garlic was the bud differentiation period, and then the classification accuracy tended to be stable.

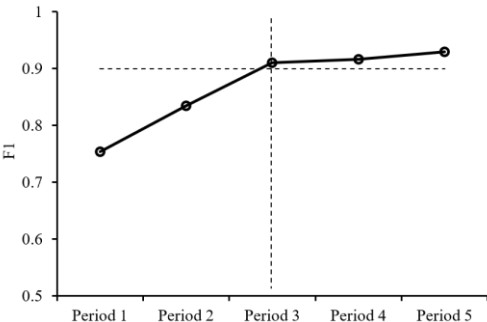

**Figure 5.** Variations in F1 of garlic. "Period 1", "Period 2", "Period 3", "Period 4", and "Period 5" represent the germination period, seedling period, bud differentiation period, elongation period, and expansion period, respectively.

### 3.3. Early-Season Garlic Distribution Map

Figure 6 shows the map of garlic distribution obtained using only 17 images of the germination, seedling, and bud differentiation periods; that is, we could obtain information on garlic planting three months before harvest. Garlic fields were mainly located in the central and northern parts of the study area, where garlic has a long history of planting and a stable market has developed. The southern part of the study area was primarily winter wheat fields, mainly due to the lack of planting experience and the lack of garlic market.

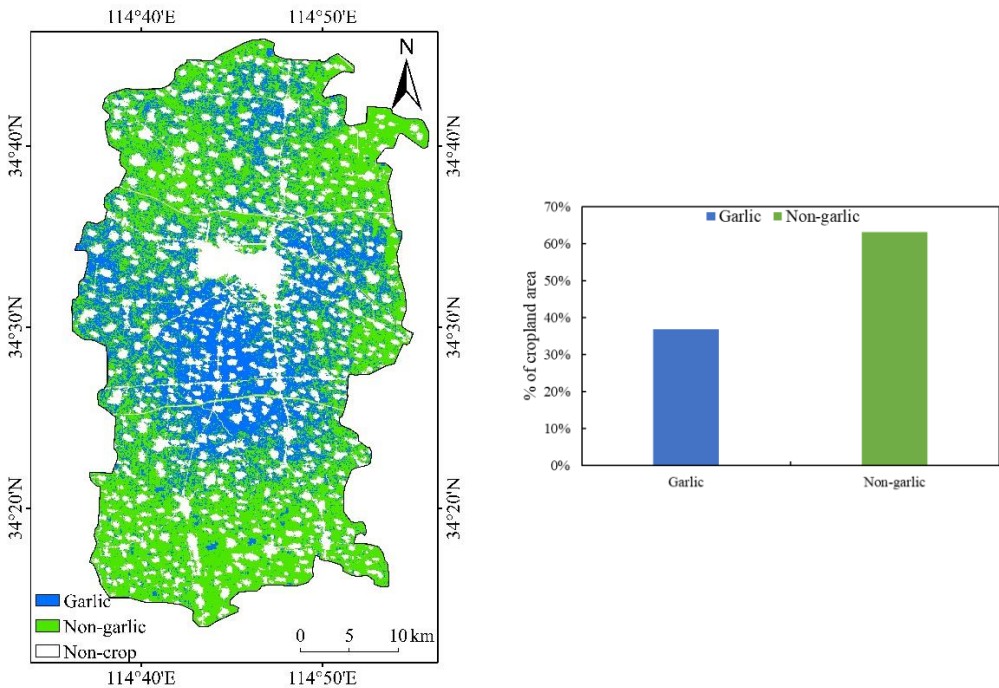

**Figure 6.** Early-season garlic distribution map.

## 4. Discussion

### 4.1. Optimal Identification Strategy

Vegetation indices are the most used metrics in crop identification. Generally, the identification of target crops is achieved by analyzing vegetation indices characteristics using machine learning or rule-based algorithms [19]. Many efforts have been devoted to exploring the importance of vegetation indices in the classification of different crops [67–69]. However, it remains unclear at which specific phenological period of the crop the vegetation index works. Here, we quantitatively analyzed the performance of vegetation indices in each phenophase using separability tests to determine the optimal performing garlic classification indices in each phenophase and to obtain the optimal identification strategy

for garlic. It is worth noting that MNDWI outperformed the other indices significantly in the early season of garlic (Figure 4), which may be attributed to its use of the SWIR band. In the early season of garlic, leaf tissue moisture content is lower than in other non-garlic crops (e.g., winter wheat), which is an important reason why SWIR plays a key role, as evidenced by the conclusions of other studies [16].

To illustrate the adaptability of the method in this study to other regions, we obtained the seasonal changes of garlic fields in other regions. Taking NDVI as an example, the seasonal changes are very similar in different regions, for example, all capturing NDVI peak in April to May (Figure 7). This demonstrates that the optimal strategy proposed in this study is equally applicable to these areas. Overall, the migration of the optimal identification strategy is feasible in the other regions with similar latitudes, but in areas with large latitudinal differences, the influence of temperature on the seasonal change of garlic should be considered. That is exactly what we perform next.

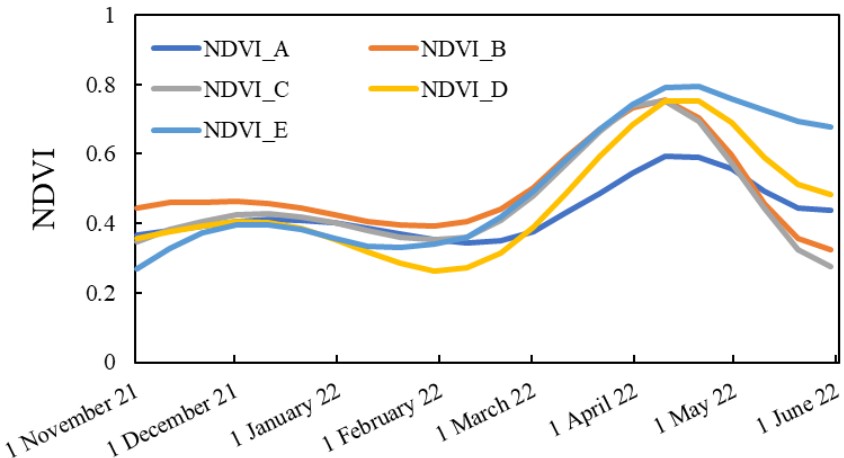

**Figure 7.** Seasonal changes of NDVI for garlic in different regions. Among them, NDVI_A, NDVI_B, NDVI_C, NDVI_D, and NDVI_E are the garlic fields in Zhengzhou, Kaifeng, Jining, Linyi, and Xuzhou, respectively.

*4.2. Earliest Identifiable Phenophase*

The accuracy of crop distribution maps increases with the number of images used [42], but this will be limited by the ease of data acquisition and the memory requirements for data processing. This implies a tradeoff between map accuracy and data amount. In this study, we tried to map the garlic distribution using early and mid-phenophase images. The automatic mapping framework proposed in this study achieved garlic identification using images from only three phenophases (germination period, seedling period, and bud differentiation period). Therefore, the earliest identifiable phenophase of garlic is the bud differentiation period. During this period, garlic starts to bolt, leaves gradually turn yellow, leaf water content and chlorophyll content keep decreasing, and the NDVI, GNDVI, and OSAVI are lower than other crops.

Further, we generated a post-season map using all available images of garlic through-out the entire lifecycle and compared the accuracy with the early-season map (Figure 8). Compared to the post-season maps, the F1, overall accuracy (OA), PA, and Kappa coefficient of the early map decreased by 0.01, and the UA was not changed. Although the accuracy of the early-season maps is limited by the number of images used [70], Figure 7 shows that the difference in accuracy is not significant compared to the post-season maps. This insignificant difference precisely demonstrates that the garlic early mapping framework proposed in this study has satisfactory accuracy while reducing data costs.

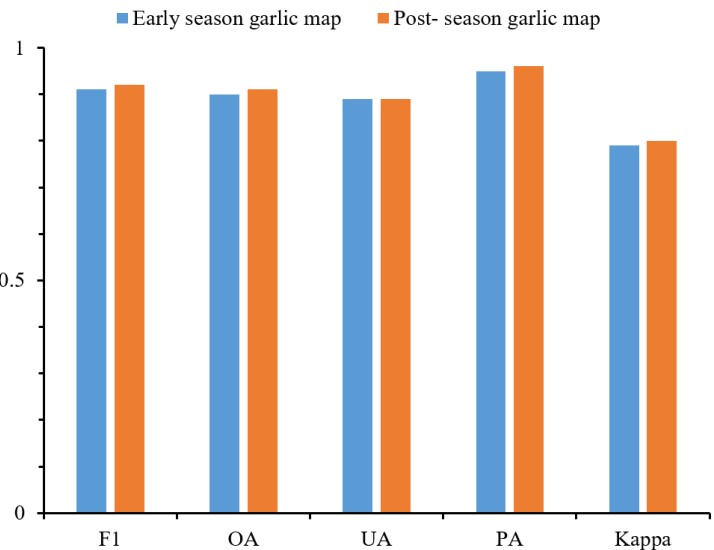

**Figure 8.** Comparison of accuracy between early-season map and post-season map.

### 4.3. Uncertainty

Several factors may affect the classification results of this study. First, reliable land cover data is an important factor to improve the accuracy of the map. The GlobeLand30 product was used with a 30 m resolution to distinguish cropland from other land types. However, there were some classification errors in this product, especially for the staggered distribution zone of cropland and villages. In addition, the product reflects the land cover in 2020, which is inconsistent with the study year (2022). These errors are easily propagated to the final garlic distribution map output. The publication of more reliable land cover data is expected to further improve the accuracy of the research results. Second, for the missing values in the VIs time series, linear interpolation was used based on neighboring pixels. When good-quality observations are missing at the peaks (valleys), the interpolation cannot reflect the real positions of crop growth [64]. In addition, linear interpolation has difficulty reflecting the long-term gaps in the data [71]. Other spatial interpolation methods such as bilinear interpolation [72] and kriging interpolation [73,74] may better fill the missing pixels, and the comparison of different interpolation methods is one of the directions of the subsequent work.

Higher-resolution data may present different results. For this, we compared the results using Sentinel-2 and Landsat 7/8 (no Landsat 9 data were available for 2020 and 2021) (Figure 9). The results show that there is almost no difference in accuracy between these two data, and the Landsat data are slightly more accurate. Therefore, the increased resolution exhibited very limited contribution to the results, and the Landsat data are enough for identifying the planting area of garlic. In addition, Landsat data of long time series provide an opportunity to explore the interannual variability of garlic fields by extending the method proposed in this study to multiple years, which is a limitation of Sentinel data. Finally, non-garlic crops include winter wheat, winter canola, and winter vegetables, and the differences between them were not discussed in this study. A comprehensive analysis of image characteristics of different types of winter crops and improving the universality of the algorithm proposed in this study are the directions of further research.

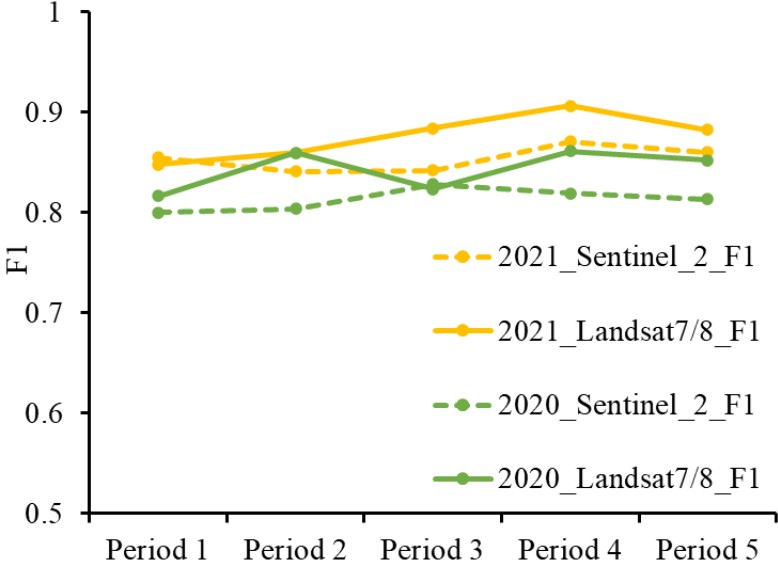

**Figure 9.** Comparison of accuracy between Sentinel-2 and Landsat 7/8 data.

## 5. Conclusions

In this study, we proposed an automatic mapping framework to identify and map garlic distribution using the time series Landsat 8/9 images. Specifically, we tested the separability of multiple vegetation indices and determined the optimal identification strategy of garlic. Further, based on this identification strategy and garlic phenology, we determined the earliest identifiable phenophase of garlic and generated the early-season garlic distribution map. Garlic can be identified earliest at the bud differentiation period with an F1 of 0.91. The garlic distribution map obtained in this study can spatially characterize the population dynamics of garlic, and apply this information to plan the precise garlic management. The garlic early-season identification algorithm proposed in this study can map garlic at the bud differentiation period, which helps government policymakers, market traders, and farmers access timely and effective garlic distribution information. Moreover, the proposed phenology-based garlic mapping algorithm framework can also be applied for mapping garlic distribution in other years or other places.

**Author Contributions:** Conceptualization, H.X.; methodology, Y.G. and H.X.; validation, Y.G., X.Z. and L.Q.; formal analysis, Y.G. and H.X.; investigation, Y.G., X.Z. and L.Q.; resources, Y.G.; data curation, Y.G. and X.Z.; writing—original draft preparation, Y.G.; writing—review and editing, H.X. and Y.Q.; visualization, Y.G.; funding acquisition, H.X. All authors have read and agreed to the published version of the manuscript.

**Funding:** This research was funded by Henan Provincial Department of Science and Technology Research Project, grant number 212102310019.

**Data Availability Statement:** Landsat 8/9 data are openly available via the Google Earth Engine.

**Acknowledgments:** We are grateful to the anonymous reviewers whose constructive suggestions have improved the quality of this study, and wish to express our gratitude to the USGS and GEE platforms for supplying Landsat data, and to "Dabieshan National Observation and Research Field Station of Forest Ecosystem at Henan" for supplying equipment.

**Conflicts of Interest:** The authors declare no conflict of interest.

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
