# Peer review of "Estimate the Earliest Phenophase for Garlic Mapping Using Time Series Landsat 8/9 Images"

_remotesensing, doi:10.3390/rs14184476_

Round 1

Reviewer 1 Report

This manuscript presents a framework that exploits Landsat 8 and 9 data in order to map early phenological stages of garlic in China. To do so, authors make use of a Random Forest on GEE and explore different radiometric indices compared to actual ground truth data for the different phenological stages. Though the idea and application are interesting, authors failed at presenting it on a proper way. Authors lack of proper scientific knowledge of different basic terms and make erroneous use of them along the manuscript. Justifications on type of data used are not truth and other sensor would work better for their task. The methodology is interesting, but authors do not exploit it on a proper way. Results are poorly presented to non existent, and the same happens with the discussion. In consequence, conclusions are not actually supported by presented results. Further details can be found in the next.

ABSTRACT

- Please add the exact information about the earliest time of identification for garlic.

- Add information about other methods to compare with, even if they are not able to identify garlic as early as you do.

INTRODUCTION

- L7. WorldView-2 is considered as Very High spatial Resolution sensor (VHR), the UVHR term is used for data under centimeter such as the one acquired by UAV/drones.

- L66. What is the average area of a crop field in China?. Why not to use Sentinel-2 (with 10m spatial resolution and higher temporal one) instead?.

- L71. Using Sentinel-2 A and B you can have higher temporal resolution, and also higher spatial resolution. Together with thei spectral bands for vegetation, you could further exploit phenological information.

- General: Some more details about methods available for exploring and or deriving phenological parameters is missing. Some examples are: (a) Y. T. Solano-Correa, F. Bovolo, and L. Bruzzone and D. Fernández-Prieto, “Automatic Derivation of Cropland Phenological Parameters by Adaptive Non-Parametric Regression of Sentinel-2 NDVI Time Series,” in 2018 IEEE International Geoscience and Remote Sensing Symposium (IGARSS), 2018, pp. 1946–1949 and (b) Maleki, M.; Arriga, N.; Barrios, J.M.; Wieneke, S.; Liu, Q.; Peñuelas, J.; Janssens, I.A.; Balzarolo, M. Estimation of Gross Primary Productivity (GPP) Phenology of a Short-Rotation Plantation Using Remotely Sensed Indices Derived from Sentinel-2 Images. Remote Sens. 2020, 12, 2104. Please cite them and search for other relevant papers in the same direction. Please be careful about the use of technical terms, specially regarding resolutions. Be aware that you need to add a discussion on the topic Landsat vs Sentinel-2. I expect to see a comparison of using the two types of sensors in order to actually determine which one works better for your goal.

MATERIALS AND METHODS

- Figure 1. the map on the left is not quiet clear. It is not possible to understand what is what and where is the location of the studied place. Please improve.

- L101. You make it sound as if the older sensors could not work well. This is not about the atmospheric corrections, but about the radiometric resolution of the sensor, which is actually  different from other existing sensors. 

- L108. How did you determine this spatial resolution?.

- L112. I have my doubts regarding how easily can you actually distinguish garlic at 30m spatial resolution. You should write down something regarding why would it be possible to detect it, and why on early stages.

- Figure 2. I see green and organge colors. Where are the yellow ones?. Please add the legend of colors directly on the figures.

- Section 2.2.3. If you were to use Sentinel-2 instead, you could also make use of the World Dynamic map recently made available on GEE for all the world and all the dates.

- Section 2.3.1. Please summarize all of this information on a table with corresponding references.

- Section 2.3.3. Other more relevant literature can be found on: (a) Misra, G.; Cawkwell, F.; Wingler, A. Status of Phenological Research Using Sentinel-2 Data: A Review. Remote Sens. 2020, 12, 2760. and (b) A Garioud, S Valero, S Giordano, C Mallet, "Recurrent-based regression of Sentinel time series for continuous vegetation monitoring," Remote Sensing of Environment, vol.263, no. 15, Sep. 2021, 112419.

- General: You are missing to add a comparison across Landsat and Sentinel-2 or to write a proper paragraph that actually justifies why do you prefer to use Landsat instead of Sentinel-2.

RESULTS

- You mentione a lot of steps and information on your method and then your results are barely two pages. What is a possible reader suppose to understand from this?. What about some further analysis with different time series lenghts?. What about comparing among two types of sensors?. What about testing for a different year?. Several things are missing.

DISCUSSION

- Section 4.1. What is the goal of this section?. I do not see any contribution.

- Figure 7. I do not see a significant difference between the two cases, but this is positive and you are not discussing it on a proper way.

- Please joing results and discussion and expand all the results and experiments that you have carried out.

- L293. The World Dynamic map is able to produce a map for each available Sentinel-2 image on GEE, you could easily use (and must do so) this map in order to properly mask the agricultural areas with the corresponding year.

- L296. If you take a look to the papers suggested for time series reconstruction, you will notice that a linear interpolation is not the best to use in order to perform this task. This denotes the lack of clear understanding of literature from authors.

- General: Since you have not presented a detailed list of results and analysis, it is difficult to actually write down a discussion section. You need to improve your experiments before considering the manuscript for publication. Once you improve the experiments, the whole mansucript will gain much more value and you will be able to get results that are more reliable for your final goal.

CONCLUSIONS

- L305. Where are these tests?.

- L308. What about presenting the maps and their corresponding accuracies for different stages?. Are the values so different?. 

- The results and experiments that you have presented so far are not enough to support these conclusions.

REFERENCES

- They do not cover the actual state of the art and a lot of basic information is missing. Authors need to perform a better analysis of literature that also helps them to make better use of scientific terms.

Reviewer 2 Report

Overall, this work is interesting. But I have some major concerns to be addressed when revising the MS as follows:

Lines 60-65 just give a very brief explanation on the methodological development of generating early or middle season crop products,without presenting a clear picture regarding the detailed methodologies, which undermines the judgement of the novelty of the current framework. So, I do recommend the authors to or-organize the methodological review substantially to give a clear clue on the method progress or advancement

Line 63  that seems to be deleted.

Line 68  releasecan be changed as launch

Line 70 publishedseems to be replaced by  provided or available

Line 75 country--county??

Notes in Figure 2 (f),(g)and (h) should be translated into English and make them more clear.And another key point is that all the data collection dates should be clearly presented because they dates can provide important phenological information of garlic growth

Line 129 access should be accessed  

Line 162, symbols ρshould be re-edited to make sure they are in agreement with those in equations 1 to 9

Equation (10): before calculating your M value, you must verify the normality of your two category samples values.If the normality is not the case, your equation 10 is invalid.

Line 169-170 Can you minimize the co-relation the three selected metrics to support your classification??

Figure 6: A legend, non-crop for the white area of the study area was missing!

Lines 264-265,If we change your research area to another biome or climatic zone (may with higher complexity of crop coexistence, such as several corps or vegetation rather than two types in your specific research site), is your optimal identification strategy truly working? This point must be deeply discussed to convince readers.  

Reviewer 3 Report

   The manuscript by Guo et al. is devoted to an important applied problem: development of methods of identification of garlic on basis of analysis of satellite images. The work seems to be interesting and perspective: however, several technical points are not clear for me.

   1. Sections 2.3.2 and 3.1: M should show efficiency of separation between groups. However, it is not clear: What groups were used for separation? It is probable that garlic and other plants (what plants? Why these plants?) were separated. However, this point should be strongly clarified because it is basis of further results.

   2. Section 2.3.2: Why M=1 was used as the separation value? Why M was calculated as abs(μ1- μ2)/(σ1+ σ2)? It should be clarified. Considering statistical approaches, other ways can be used. E.g., total σ of two croups can be calculated as ((σ1^2+ σ2^2)^0.5); tα = abs(μ1- μ2)/σ. Using Student’s test, we can selected tα for specific level of significance of difference (e.g., p < 0.05).

   3. Sections 2.3.2 and 3.1: What criteria of selection of three reflectance indices (the best indices?) for each phenophase were used? E.g., Figure 4 shows that at least five indices had M>1 at the elongation period; in contrast, only MNDWI had M>1 in the most of points of seedling period. It should be clarified.

   4.  Sections 2.3.3, 2.3.4, and 3.2: Transition from vegetation indices to classification and calculation of F1 should be described in more detail. Now, this part of the manuscript is not clear.

   5. Section 2.3.1: Criteria of selection of used vegetation indices should be described in more detail. Additionally, I suppose that these indices and equations should be presented as Table because this presentation seems to be more suitable for potential readers

Round 2

Reviewer 1 Report

- I appreciate the effort from authors in orther to improve the manuscript's quality. It has indeed been improved in several aspects, but the most critical part still remains unchanged in a proper manner. This is regarding the lack of proper experimental designs and better justification on their selection. You can find some details below.

- L125-127. This is still a bit unprecised. Please check "F. Pacifici, N. Longbotham and W. J. Emery, "The Importance of Physical Quantities for the Analysis of Multitemporal and Multiangular Optical Very High Spatial Resolution Images," in IEEE Transactions on Geoscience and Remote Sensing, vol. 52, no. 10, pp. 6241-6256, Oct. 2014, doi: 10.1109/TGRS.2013.2295819." For better understanding.

- L176. "presented in Table 1"

- Results and discussion are still presented as separate section and are still lacking proper amount of results to support their different claims. While authors have replied and justify the reasons why Sentinel-2 is not preferred, I am still not convinced about this, since I cannot see any particular results supporting this. Several things were said on the replies to my previous review, but they have not been added to the revised manuscript. Authors still need to add a proper experimental design and analysis that allows the reader to arrive to the conclusions presented. An analysis regarding suitability of spatial resolution for analyzing the small crop fields (0.2ha) in China is required. It is impossible to believe that this is the first manuscript reporting that Landsat works better than Sentinel-2 for analyzing information at such a small scale. Specially while clearly claiming a possible error on Sentinel-2 data as the justification on why is it not working for their data. Without a proper set of experiments, analysis and results, I cannot recommend this manuscript for publication.

- Please show a histogram distribution of garlic crop fields over the studied area.

Reviewer 2 Report

Thanks for the authors' substantial efforts to refine the MS. All of my concerns have been adequately addressed in the revised version. I recommend its publication in the current form.

Reviewer 3 Report

Authors considered my comments. I suppose that this interesting work can be accepted.